The acute effects of simulated hypoxic training at different altitudes on oxidative stress and muscle damage in elite long-distance runners

Sarikaya Mücahit mucahitsarikaya@yyu.edu.tr 1
Öğe Beyza 2
Embiyaoğlu Nuri Mert 1
Selçuk Muzaffer 1
Çınar Vedat 3
Öner Salih 1
Gencer Yıldırım Gökhan 4
Aslan Mehdi 5
Ulema Mustafa Sencer 1
Yarayan Yunus Emre 5
Keskin Kadir 6
Alkhamees Nouf H. 7
Sheeha Bodor Bin 7
Grivas Gerasimos V. 8
AL-Mhanna Sameer Badri 9
Batrakoulis Alexios abatrako@phyed.duth.gr alexis_batrakoulis_75@hotmail.com 10 11
1 School of Physical Education and Sports, Department of Physical Education and Sports Teaching, Van Yüzüncü Yıl University , Van , Turkey
2 School of Physical Education and Sports, Department of Coaching Education, Van Yüzüncü Yıl University , Van , Turkey
3 Faculty of Sport Science, Department of Physical Education and Sports Teaching, Fırat Üniversitesi , Elâzığ , Turkey
4 Faculty of Sport Science, Department of Coaching Education, Mersin University , Mersin , Turkey
5 School of Physical Education and Sports, Department of Coaching Education, Siirt University , Siirt , Turkey
6 Faculty of Sport Science, Department of Physical Education and Sports Teaching, Gazi University , Ankara , Turkey
7 Department of Rehabilitation, College of Health and Rehabilitation Sciences, Princess Nourah bint Abdulrahman University , Riyadh , Saudi Arabia
8 Physical Education and Sports, Division of Humanities and Political Sciences, Hellenic Naval Academy , Athens , Piraeus , Greece
9 Department of Physiology, School of Medical Sciences, Universiti Sains Malaysia, Kubang Kerian , Kelantan , Malaysia
10 Department of Physical Education and Sport Science, Democritus University of Thrace , Komotini , Greece
11 Department of Physical Education and Sport Science, University of Thessaly , Trikala , Greece
Vandoni Matteo
Electronic publication date: 2025 May 12
Publication date: 2025
Volume: 13
Electronic Location ID: e19338
Received 2024 Dec 10; Accepted 2025 Mar 27
Copyright: ©2025 Sarikaya et al.
Copyright year: 2025
Copyright holder: Sarikaya et al.
License: This is an open access article distributed under the terms of the Creative Commons Attribution License, which permits unrestricted use, distribution, reproduction and adaptation in any medium and for any purpose provided that it is properly attributed. For attribution, the original author(s), title, publication source (PeerJ) and either DOI or URL of the article must be cited.
License URL: https://creativecommons.org/licenses/by/4.0/

Keywords: Hypoxia, Training, Oxidative stress, Muscle damage, Altitude

Funding: Princess Nourah bint Abdulrahman University Researchers Supporting Project PNURSP2025R422 Princess Nourah bint Abdulrahman University, Riyadh, Saudi Arabia This research was funded by Princess Nourah bint Abdulrahman University Researchers Supporting Project number (PNURSP2025R422), Princess Nourah bint Abdulrahman University, Riyadh, Saudi Arabia. The funders had no role in study design, data collection and analysis, decision to publish, or preparation of the manuscript.

==============================
Background

Understanding the impact of altitude on muscle damage and oxidative stress is essential for optimizing training and recovery strategies for athletes exposed to high-altitude conditions. Therefore, this study aimed to investigate the effects of acute exercise at different altitudes on oxidative stress and muscle damage.

Methods

A total of twelve elite long-distance runners (mean age: 20.3 ± 1.5 years) from different branches participated in the study. The exercise protocol was the Bruce submaximal treadmill exercise test, which was conducted under three simulated hypoxic conditions (at 1,700 m, 2,450 m, and 3,200 m) and one normoxic condition (sea level). All measurements took place at the same time of the day. After the exercise protocol, 5 ml venous blood samples were taken from the participants, while heart rate and oxygen saturation were monitored at the 3rd, 6th, 9th, and 12th minutes during the exercise.

Results

Significant altitude-dependent variations were observed in oxidative stress markers, with total oxidant status (TOS) (p = 0.017) and malondialdehyde (MDA) (p < 0.001) levels increasing at higher altitudes, while total antioxidant status (TAS) (p < 0.001) exhibited an elevation and oxidative stress index (OSI) (p < 0.001) demonstrated a decline as altitude increased. However, no significant difference was found in creatine kinase (CK, p = 0.059) levels. Additionally, there were significant differences in the oxygen saturation measurement taken at the 3rd (p < 0.001), 6th (p < 0.001), 9th (p < 0.001), and 12th (p < 0.001), minutes following the exercise session. There was no difference in the pulse measurement taken at the 3rd and 12th minutes, but a difference was observed at the 6th and 9th minutes post-exercise (p < 0.01).

Conclusions

In conclusion, the study determined that endurance exercises performed under simulated normobaric hypoxia at different altitudes increased TAS and reduced OSI in elite long-distance runners. The increase in TAS and the reduction in OSI were more pronounced at higher altitudes, particularly at 2,450 m and 3,200 m, compared to sea level. These findings highlight the need for altitude-specific training and recovery strategies to minimize oxidative stress and muscle damage in athletes.

Introduction

The metabolic effects and physiological responses of exercise primarily depend on the biochemical reactions occurring in muscle cells (Hargreaves & Spriet, 2020). However, the physiological stress induced by exercise can also lead to side effects such as muscle damage and oxidative stress (Powers & Jackson, 2008; Quindry et al., 2016). Oxidative stress has long been a hotspot of exercise-based research (Finaud, Lac & Filaire, 2006; Kruk et al., 2022). In recent years, intensive research has been conducted to understand oxidative stress and the physiological responses of exercise under different conditions, including high altitude (Quindry et al., 2016; Li et al., 2024). Adaptations to altitude progressively increase at elevations starting from approximately 1,000–1,500 m, primarily due to reduced oxygen availability and the subsequent physiological compensations, such as increased erythropoiesis and ventilatory responses (Miller et al., 2013; Lukanova-Jakubowska et al., 2022). High-altitude adaptation is driven by various physiological and molecular mechanisms in response to hypoxia (Pham, Parikh & Heinrich, 2021). The body compensates for reduced oxygen availability by increasing breathing and heart rate, while boosting erythrocyte production to enhance oxygen transport. On a molecular level, transcription factors like HIF-1 regulate energy metabolism and trigger antioxidant defenses to combat oxidative stress (Pham, Parikh & Heinrich, 2021). These adaptations are crucial for both enhancing physical performance and preventing altitude-related illnesses (Mallet et al., 2023; Vignati et al., 2023). Current knowledge about overexertion and oxidative stress at high altitudes is generally derived from a limited number of field studies. Alongside field studies, well-controlled laboratory studies on exercise performance and blood biomarker changes in humans artificially exposed to hypoxia also contribute significantly to this field (Gore, Clark & Saunders, 2007; Millet et al., 2010; Millet, Faiss & Pialoux, 2012; Karayigit et al., 2022).

Exposure to high altitude, which is associated with decreased oxygen pressure, can cause oxidative/reductive stress, increased formation of reactive oxygen and nitrogen species (RONS), and associated oxidative damage to lipids, proteins, and DNA (Dosek et al., 2007; Powers & Jackson, 2008). Various RONS-producing systems are activated during high-altitude exposure, including the mitochondrial electron transport chain, xanthine oxidase, and nitric oxide synthase. High altitude appears to weaken both enzymatic and non-enzymatic antioxidant systems (Chao et al., 1999; Radak et al., 2014). The pattern of oxidative damage associated with high altitude exposure is similar to ischemia/reperfusion injury. This adaptive process to the oxidative challenge requires a relatively long period (Cadenas, 2018; Zhang et al., 2019; Mallet et al., 2023). Physical exercise or increased physical activity at high altitudes may exacerbate the extent of oxidative threat (Maiti et al., 2006). The fact that acute exercise induces oxidative cell damage and contributes to systemic oxidative stress was first established by Dillard et al. (1978). Up-to-date investigators have clearly shown that acute exercise training increases oxidative stress levels predominantly with the skeletal muscle and blood (Powers, Smuder & Criswell, 2011; Powers, Radak & Ji, 2016). In this context, it is important to examine the effects of exercise performed at different altitudes on creatine kinase (CK) levels and oxidative stress (Soliman et al., 2022). Determining the relationship between muscle damage and high altitude will provide important information for athletes living and exercising at high altitudes, for athletes and coaches performing high-altitude camps, and for optimizing training methods and recovery processes. The selected biomarkers—heart rate, blood oxygen saturation, total antioxidant status (TAS), total oxidant status (TOS), malondialdehyde (MDA), and creatine kinase-MM (CK-MM)—are widely recognized for assessing oxidative stress and muscle damage under hypoxic conditions (Brancaccio, Maffulli & Limongelli, 2007; Mazzeo, 2008; Kong et al., 2022). Given these biomarkers, different altitudes may have varying effects on exercise performance and physiological responses in both hypobaric and normobaric conditions (Wilber, 2007; Girard et al., 2020). This study aims to investigate the acute effects of exercises performed at different altitudes on CK levels and oxidative stress in elite long-distance runners. It was hypothesized that acute hypoxic exercise at different altitudes would significantly influence oxidative stress markers, creatine kinase levels, and cardiovascular responses.

Materials and Methods

Participants

To determine the appropriate sample size for this study, a power analysis was conducted using G*Power software for Repeated Measures ANOVA (1-β power = 80%, alpha = 0.05, effect size = 0.40, correlation value = 0.7). It was calculated that a minimum of 8 participants would be sufficient to achieve adequate statistical power (Cohen, 1992). To account for potential withdrawals or missing data, the final sample size was increased to 12 participants, as recommended by Suresh & Chandrashekara (2012).

This study included 12 elite long-distance runners (age: 20.3 ± 1.5 years, body weight: 63.5 ± 15.2 kg, height: 171.8 ± 7.9 cm, body mass index: 21.4 ± 4.3, resting heart rate: 49.0 ± 3.0 bpm) residing in Van province of Turkey at an altitude of 1,700 m. To ensure eligibility, participants’ health status and physical activity levels were assessed using a health history questionnaire. Inclusion criteria required that all athletes (1) were free from chronic illness, (2) had no musculoskeletal injuries for at least 6 months prior to the study, and (3) had no history of acute mountain sickness. The physical and physiological characteristics of the study group are shown in Table 1.

Table 1 The descriptive characteristics of the study participants (n = 12), including age, body weight, height, body mass index (BMI), and resting heart rate.

Values are expressed as mean ± standard deviation (SD), with the minimum and maximum values provided for each variable.

Variables	n	Average ± SD	Min	Max	
Age (years)	12	20.3 ± 1.5	18	23	
Body Weight (kg)	12	63.5 ± 15.2	51.0	98.5	
Height (cm)	12	171.8 ± 7.9	155.5	184.0	
BMI (kg/m2)	12	21.4 ± 4.3	17.70	32.00	
Resting Heart Rate (bpm)	12	49.0 ± 3.0	45	52	

Additionally, the elite status of the athletes was objectively determined using the International Association of Athletics Federations (IAAF) scoring tables, based on their season-best performances from 2022 to 2023. Only those classified as “high-level” according to the IAAF criteria were included (Spiriev, 2017). This selection ensured that all participants met the competitive standards required for national and international participation. Participants were fully informed about the risks, benefits, and procedures of the study, and written informed consent was obtained. The study was conducted in accordance with the Declaration of Helsinki and approved by the Ethics Committee of Van Yüzüncü Yıl University (protocol code TYD-2021-9553, approval date: 15 May 2021).

Study design

Participants exercised under four different conditions: normoxic conditions (sea level) and normobaric altitudes of 1,700 m, 2,450 m, and 3,200 m. No acclimatization was required, as the participants had been living at an altitude of 1,700 m for an extended period. The study was conducted using a single-blind design, meaning that participants were unaware of the altitude condition they were exposed to during each trial to minimize potential psychological or physiological bias. The altitude conditions were randomized, ensuring that the order in which participants experienced different altitudes was assigned randomly to prevent order effects or adaptation biases. A crossover design was implemented, where each participant underwent all altitude conditions, allowing for within-subject comparisons and reducing inter-individual variability. The study was also balanced, ensuring that each altitude condition was tested an equal number of times across participants, thus maintaining uniform distribution and preventing overrepresentation of any specific condition.

A quantitative research method was used, and the study followed a single-group, four-treatment time series design, which is one of the quasi-experimental research models (Gliner, Morgan & Leech, 2017). Each exercise session at a different altitude was separated by at least seven days to mitigate carryover effects. The exercise protocol followed the Bruce submaximal treadmill test under the four simulated normobaric hypoxic conditions (1,700 m, 2,450 m, and 3,200 m) and normoxic conditions (sea level). Since severe hypoxia (>3,000 m) can cause adverse health effects such as high-altitude sickness, the hypoxic levels selected in this study were based on safety and practical considerations. This approach ensured that adequate metabolic stress was induced while minimizing the risk to participants (Millet et al., 2016; Strom et al., 2018; Burtscher et al., 2022).

Before the study, participants provided detailed information about their alcohol, caffeine, and smoking habits, as well as their use of medications and ergogenic aids, through a specially designed structured form aimed at assessing lifestyle factors that could influence oxidative stress levels. Additionally, they were instructed to follow a standardized diet starting three days before the first measurement, consisting of balanced meals with controlled portions of carbohydrates, proteins, and fats. This diet was designed to control the effects of consumed foods on antioxidant levels and prevent data distortion. To ensure consistency, each measurement was conducted at seven-day intervals and at the same time throughout the study. All exercise sessions were conducted in a Hypoxico Everest Summit II Altitude Generator, with temperature and relative humidity maintained at 20 °C and 40%, respectively.

Measurements

Prior the exercise tests, a 15-minute classical warm-up protocol was applied under normoxic conditions. This warm-up included 5 min of low-intensity jogging at approximately 50–60% of maximum heart rate to gradually increase heart rate and body temperature, followed by 5 min of dynamic stretching targeting key muscle groups involved in treadmill running, such as the quadriceps, hamstrings, calves, and gluteal muscles. The final 5 min consisted of mobility drills, including high knees and leg swings, to enhance joint mobility and neuromuscular activation. Following the warm-up, participants performed the Bruce submaximal treadmill exercise test (Strom et al., 2018). Upon completion of the protocol, a five mL venous blood sample was collected.

Bruce submaximal treadmill exercise test

The Bruce submaximal treadmill exercise test lasted a total of 12 min, consisting of four progressive stages designed to impose a standardized and controlled workload. In the 1st stage, participants walked for 3 min at a speed of 1.7 mph with a 10% incline. In the 2nd stage, they walked for 3 min at a speed of 2.5 mph with a 12% incline. In the 3rd stage, the exercise continued for 3 min at a speed of 3.4 mph with a 14% incline. In the 4th and final stage, participants walked for another 3 min at a speed of 4.2 mph with a 16% incline. The exercise protocol was kept constant for each measurement day, and the ambient altitude was set in four different ways: 1st measurement at sea level (normoxic conditions), 2nd measurement at 1,700 m, 3rd measurement at 2,450 m, and 4th measurement at 3,200 m. A Hypoxico Everest Summit II Altitude Generator was used to adjust the ambient partial pressure according to these altitudes (Harwood, Wright & Burnet, 2022). The exercise sessions were conducted using a mask system, ensuring controlled exposure to simulated altitude conditions. Additionally, the athletes’ oxygen saturation levels and heart rate variables were monitored under the supervision of a doctor throughout the exercise. This allowed for continuous real-time monitoring of physiological responses to ensure participant safety and the accuracy of data collection. To measure oxygen saturation and heart rate, the Masimo Radical-7 Pulse Oximeter and Polar H10 heart rate monitor were used. Physiological parameters were continuously monitored throughout the exercise, with data recorded every minute. Additionally, oxygen saturation (SpO2) and heart rate measurements were specifically documented at the 3rd, 6th, 9th, and 12th minutes to assess trends over time. The Masimo Radical-7 is an advanced, clinically-approved pulse oximeter with an SpO2 accuracy of ±2% for values between 70% and 100%. It can also provide accurate results even under low perfusion conditions. The Polar H10 is a chest-strap heart rate monitor that provides highly accurate ECG data.

Blood sampling and assays

A five mL venous blood sample was collected from the participants’ antecubital veins immediately after completing the Bruce submaximal treadmill exercise test under normoxic conditions and at simulated altitudes of 1,700 m, 2,450 m, and 3,200 m. Blood samples were centrifuged at 3,600 rpm at +4 °C for 5 min, and the serum was separated and stored at −20 °C until analysis at Van YYÜ Dursun Odabaş Medical Center.

Serum levels of CK-MM, malondialdehyde (MDA), total antioxidant status (TAS), and total oxidant status (TOS) were measured. CK-MM and MDA levels were determined using the ELISA method, while TAS and TOS levels were assessed via an automated colorimetric method (Erel, 2005). The oxidative stress index (OSI) was calculated as the TOS/TAS ratio to evaluate the overall oxidative-antioxidative balance (Sırmatel et al., 2007). Heart rate and arterial oxygen saturation were used to monitor cardiovascular and respiratory responses, while TAS, TOS, MDA, and CK-MM served as key indicators of oxidative stress, lipid peroxidation, and muscle response to exercise (Spirlandeli, Deminice & Jordao, 2014; Janion et al., 2022).

Statistical analysis

Data analysis was done using the SPSS 25 package program. The normality of the data was checked with the Kolmogorov–Smirnov and Shapiro–Wilk tests and the homogeneity with the Levene test. A one way repeated measures ANOVA was used to compare parameters at different altitudes. In cases where the sphericity assumption was not met, the Greenhouse-Geisser correction was used. For values with significant differences as a result of repeated measurements, the Bonferroni post hoc test was used to determine which data group caused the difference. The significance level for analysis was evaluated as p < 0.05. For altitudes where a significant difference was observed, Cohen’s d was calculated, presented with 95% confidence intervals, and classified as small (d = 0.2), medium (d = 0.5), and large (d ≥ 0.8) (Cohen, 1988).

Results

Oxidative stress markers and creatin kinase

Figures 1 and 2 summarizes the results of TAS, TOS, MDA, OSI, and CK levels across different altitudes. TAS levels were significantly different between the altitudes (p < 0.001). TAS levels were significantly higher in 1,700 m, 2,450, and 3,200 m compared to the normoxia condition (p < 0.005, d = 1.72 [0.78−2.65], p < 0.001, d = 2.94 [1.78−4.09], p < 0.001, d = 4.17 [2.74−5.60] respectively). Additionally, 3,200 m had significantly higher TAS level compared to 1,700 m and 2,450 m (p < 0.001, d = 1.66 [0.73−2.59], p < 0.05; d = 1.0 [0.15−1.85] respectively). For the TOS levels the only significant difference was detected between 2,450 m and normoxic condition (p < 0.05, d = 1.99 [1.01−2.97]. MDA levels showed significant differences across the altitudes (p < 0.001). MDA levels were higher at 2,450 m and 3,200 m compared to the normoxic condition p < 0.001, d = 1.49 [0.58−2.39], p < 0.005, d = 2.99 [1.82−4.15] respectively). OSI levels were significantly different between the altitudes (p < 0.001). OSI was significantly higher under normoxic condition compared to 1,700 m, 2,450 m, and 3,200 m (p < 0.001, d = 1.54 [0.63−2.45], p < 0.001, d = 4.10 [2.69−5.51], p < 0.001, d = 4.61 [3.08−6.14] respectively) (Fig. 1). CK levels showed no differences between the altitudes (p > 0.05) (Fig. 2).

Figure 1 (A–D) Effects of altitude on oxidative stress and antioxidant capacity in human subjects.

The effects of increasing altitude on oxidative stress and antioxidant capacity. Data are expressed as mean ± SD for different conditions: normoxia (sea level), and three hypoxic conditions at 1,700 m, 2,450 m, and 3,200 m above sea level.

Figure 2 Effect of altitude on creatine kinase (CK) levels.

The levels of creatine kinase (CK), a biomarker of muscle damage, at different altitudes: normoxia (sea level), 1,700 m, 2,450 m, and 3,200 m. Data are expressed as mean ± SD.

Variations in oxygen saturation across different intervals

Figure 3 presents the variations in oxygen saturation (OS) across different altitudes and time points (3rd, 6th, 9th, and 12th minutes). A significant altitude effect was observed at all time points (p < 0.001), with OS decreasing progressively as altitude increased.

Figure 3 Effects of altitude on oxygen saturation during exercise.

The effects of different altitudes (normoxia, 1,700 m, 2,450 m, and 3,200 m) on oxygen saturation (%) at four time points (3 min, 6 min, 9 min, and 12 min) during exercise. Data are expressed as mean ± SD.

At the 3rd minute, OS was significantly lower at 2,450 m (p < 0.001, d = 3.32 [2.08−4.55]) and 3,200 m (p < 0.001, d = 3.97 [2.59−3.35]) compared to sea level. Additionally, OS at 2,450 m was significantly lower than at 1,700 m (p < 0.005, d = 2.48 [1.41−3.54]), and 3,200 m showed significantly lower OS compared to both 1,700 m and 2,450 m (p < 0.001, d = 3.58 [2.29−4.87]; p < 0.001, d = 2.27 [1.24−3.30]).

At the 6th minute, a similar trend was observed, with OS significantly lower at 2,450 m (p < 0.001, d = 2.16 [1.15−3.17]) and 3,200 m (p < 0.001, d = 5.91 [4.05−7.76]) compared to sea level. OS at 2,450 m was also lower than at 1,700 m (p < 0.001, d = 1.85 [0.89−2.80]), while 3,200 m exhibited significantly lower OS than both 1,700 m and 2,450 m (p < 0.001, d = 5.63 [3.84−7.41]; p < 0.001, d = 3.21 [2.00−4.43]).

At the 9th minute, OS remained significantly lower at 2,450 m (p < 0.001, d = 2.90 [1.75−4.05]) and 3,200 m (p < 0.001, d = 6.42 [4.43−8.40]) compared to sea level. OS at 2,450 m was also significantly lower than at 1,700 m (p < 0.001, d = 2.38 [1.34−3.43]), and 3,200 m had significantly lower OS than both 1,700 m and 2,450 m (p < 0.001, d = 5.99 [4.11−7.86]; p < 0.001, d = 3.77 [2.44−5.11]).

At the 12th minute, OS continued to decrease significantly at 2,450 m (p < 0.001, d = 2.50 [1.43−3.57]) and 3,200 m (p < 0.001, d = 5.42 [3.69−7.15]) compared to sea level. OS at 2,450 m was lower than at 1,700 m (p < 0.001, d = 2.38 [1.33−3.43]), while 3,200 m had significantly lower OS than both 1,700 m and 2,450 m (p < 0.001, d = 5.28 [3.58−6.97]; p < 0.001, d = 2.34 [1.30−3.37]). Overall, these results indicate a progressive decline in oxygen saturation with increasing altitude and exercise duration, with the most pronounced effects observed at 3,200 m, particularly in the later stages of exercise (Fig. 3).

Changes in heart rate (beat per minute) at different altitudes

Figure 4 presents the variations in heart rate (HR) across different altitudes and time points (3rd, 6th, 9th, and 12th minutes). A significant altitude effect was observed at all time points (p < 0.001), with HR increasing progressively as altitude increased.

Figure 4 Effects of altitude on heart rate during exercise.

The effects of increasing altitude (normoxia, 1,700 m, 2,450 m, and 3,200 m) on heart rate (BPM) at four time points (3 min, 6 min, 9 min, and 12 min) during exercise. Data are expressed as mean ± SD.

At the 3rd minute, HR was significantly higher at 1,700 m (p < 0.001, d = 2.66 [1.56−3.76]), 2,450 m (p < 0.005, d = 1.51 [0.60−2.42]), and 3,200 m (p < 0.001, d = 2.84 [1.71−3.98]) compared to the normoxic condition. Additionally, HR at 3,200 m was significantly higher than at 1,700 m (p < 0.005, d = 1.31 [0.42–2.19]) and 2,450 m (p < 0.001, d = 1.43 [0.54–2.33]), while HR at 2,450 m was also significantly higher than at 1,700 m (p < 0.005, d = 0.98 [0.25–1.71]).

At the 6th minute, HR remained significantly higher at 1,700 m (p < 0.001, d = 4.13 [2.72−5.55]), 2,450 m (p < 0.001, d = 2.17 [1.16−3.18]), and 3,200 m (p < 0.001, d = 5.60 [3.82−7.38]) compared to normoxia. Additionally, HR at 3,200 m was significantly higher than at 1,700 m (p < 0.001, d = 1.31 [0.42−2.19]) and 2,450 m (p < 0.001, d = 1.43 [0.54−2.33]).

At the 9th minute, HR was significantly higher at 1,700 m (p < 0.001, d = 4.83 [3.25−6.42]), 2,450 m (p < 0.001, d = 7.73 [5.40–10.06]), and 3,200 m (p < 0.001, d = 6.03 [4.15−7.92]) compared to the normoxic condition. HR was also significantly lower at 1,700 m compared to 2,450 m (p < 0.005, d = 0.17 [−0.62−0.97]) and 3,200 m (p < 0.001, d = 1.23 [0.35−2.10]). Furthermore, HR at 3,200 m was significantly higher than at 2,450 m (p < 0.005, d = 1.27 [0.35−2.10]).

At the 12th minute, a significant difference was observed only between the normoxic condition and 3,200 m (p < 0.001, d = 1.68 [0.75−2.61]). These findings indicate a progressive increase in heart rate with increasing altitude, with the most pronounced effects observed at 3,200 m, particularly in the later stages of exercise (Fig. 4).

Discussion

While high altitude can change the oxidative stress response in different simulated altitudes, this study was conducted to determine how oxidative stress is affected under exercise conditions. The main findings of the present study indicated that TAS, TOS, MDA, and OSI levels varied significantly across different altitudes, whereas CK levels remained unchanged. In particular, TAS levels were elevated at higher altitudes, while OSI levels were lower in hypoxic conditions. On the other hand, CK levels did not show significant changes, which might be due to multiple factors, despite varying exercise intensities and altitudes. One potential explanation for the lack of significant CK elevation is the type of exercise performed in this study. CK levels typically increase following exercises that involve a high degree of eccentric contractions or prolonged, intense muscle loading. However, the Bruce submaximal treadmill protocol primarily involves concentric contractions, which are less likely to induce muscle damage compared to eccentric movements such as downhill running or high-intensity resistance training (Clarkson & Hubal, 2002; Latham et al., 2008).

Additionally, long-distance runners, who formed the sample group, often develop a degree of physiological adaptation to frequent high-intensity training. These adaptations reduce CK response to exercise by enhancing muscle repair mechanisms and limiting muscle fiber damage under conditions that might otherwise elevate CK in untrained individuals. This resilience in elite athletes can attenuate CK fluctuations even when exercise is performed at higher altitudes, where muscle oxygenation might be more compromised (Brancaccio, Maffulli & Limongelli, 2007; Scalco et al., 2016). The possible lack of significant changes in MDA levels can be explained by various factors. Oxidative stress occurring during high-altitude exercise might be controlled by adaptive antioxidant defense mechanisms, which are often highly developed in elite endurance athletes. These adaptations increase resilience to oxidative stress and can prevent notable elevations in lipid peroxidation markers like MDA. Since MDA typically rises in response to heightened cellular damage, the robust antioxidant systems in such athletes might limit oxidative damage and, consequently, help stabilize MDA levels (Pialoux et al., 2006; Furian, Tannheimer & Burtscher, 2022). Additionally, low-oxygen environments reduce muscle oxygenation, which can further trigger oxidative stress. However, endurance athletes who are adapted to high altitudes can maintain cellular homeostasis by upregulating antioxidant enzyme activities, effectively minimizing oxidative damage. This adaptive process helps counteract the harmful effects of free radicals and restricts shifts in MDA levels (Chapman, 2013).

León-López et al. (2018) investigated the effects of sea level and high-altitude training on oxidative stress and antioxidant enzyme activities in professional swimmers. They reported that high-altitude training led to an increase in markers of oxidative stress, such as protein oxidation (AOPP) and MDA, indicating elevated lipid peroxidation. However, they observed a concomitant increase in antioxidant enzyme activities, suggesting a compensatory response to alleviate oxidative damage. Similarly, Belviranlıet al. (2017) examined the effect of high-altitude training on oxidative stress and antioxidant defense markers in pentathlon athletes. They reported that altitude training increased oxidative stress assessed by MDA, a marker of lipid peroxidation, and non-enzymatic antioxidant levels measured by reduced glutathione (GSH). At the same time, it did not affect enzymatic antioxidant levels assessed by superoxide dismutase (SOD) activity. Interestingly, while some studies have reported an increase in oxidative stress markers at high altitudes, others have suggested that well-adapted athletes exhibit a blunted oxidative stress response. For example, Mallet et al. (2023) observed increased oxidative stress following prolonged high-altitude training, whereas (Dosek et al., 2007) found that short-term hypoxic exposure in trained endurance athletes did not elicit the same response. These differences highlight the role of training background and exposure duration in oxidative stress regulation.

Dosek et al. (2007) stated that oxidative stress, especially when induced by hypobaric hypoxia, causes structural changes and cellular damage in lipids, proteins, and DNA. Hypoxia can occur in two different forms: normobaric and hypobaric. Hypobaric hypoxia is characterized by reduced barometric pressure at high altitudes, whereas normobaric hypoxia occurs when oxygen concentration is reduced while barometric pressure remains constant at sea level (Dosek et al., 2007). Debevec, Millet & Pialoux (2017) examined hypoxia-induced oxidative stress modeling with physical activity. They found that hypobaric hypoxia induces oxidative stress, as indicated by changes in oxidative stress markers. However, they also observed an upregulation of antioxidant defense mechanisms, suggesting an adaptive response to counteract oxidative damage. The results of our study show that TAS increases in direct proportion to altitude, while OSI decreases. This may be due to the decrease in oxygen availability caused by low atmospheric pressure (Sinex & Chapman, 2015). This lack of oxygen triggers a physiological response in the body known as hypoxia. The body adapts to hypoxic conditions by increasing antioxidant production to counteract oxidative stress, which is caused by an imbalance between free radicals and antioxidants (Dosek et al., 2007). Exercise-induced oxidative stress occurs during physical activity because metabolic processes produce free radicals. However, a limitation of this study is that oxidative stress markers were only assessed immediately post-exercise. Previous research suggests that certain biomarkers, including CK and MDA, may peak 24 to 48 h after exercise rather than immediately post-exercise (Clarkson & Hubal, 2002).

Collectively, findings from these studies (Dosek et al., 2007; Belviranlıet al., 2017; Debevec, Millet & Pialoux, 2017; León-López et al., 2018; Elmas & Elmas, 2020) suggest that high-altitude exercise can induce oxidative stress, as evidenced by changes in oxidative stress markers such as MDA. However, it is important to note that these changes are often accompanied by a simultaneous upregulation of antioxidant defense mechanisms, reflected by increased antioxidant enzyme activities and enhanced TAS. These adaptive responses may contribute to the restoration of redox balance and the overall improvement in athletic performance observed in athletes performing high-altitude exercises. It is worth noting that the interaction between oxidative stress and performance improvements in high-altitude exercise requires further investigation. The exact mechanisms underlying the relationship between oxidative stress, antioxidant responses, and athletic performance adaptations in response to high-altitude exercise have not been fully elucidated. Future studies should focus on elucidating the signaling pathways involved in adaptation to oxidative stress and identifying strategies to optimize the antioxidant defense system during high-altitude exercise.

In terms of saturation measurements, a significant difference was found between the 3rd, 6th, 9th, and 12th min. Many studies have examined the relationship between high-altitude exercise and oxygen saturation levels. Chapman et al. (2011) investigated changes in oxygen saturation during high-altitude endurance training. They observed that as athletes train at higher altitudes, oxygen saturation levels tend to decrease due to reduced oxygen availability. However, Chapman et al. (2016) indicated that athletes can adapt to high altitudes through acclimatization and adaptation processes. These adaptations include improvements in oxygen-carrying capacity and enhanced oxygen utilization efficiency. As a result, athletes may experience reduced declines in oxygen saturation levels over time as they adapt to the challenges posed by high-altitude training.

Moreover, Siebenmann et al. (2012) conducted a study examining the effects of high-altitude training on oxygen saturation and performance in endurance athletes. Initially, exposure to high altitude led to a decrease in oxygen saturation among athletes. However, with continued training and acclimatization, athletes experienced increased saturation values and improvements in performance. The results of our study indicated that saturation values decreased with increasing altitude. This decrease may be attributed to reduced ambient oxygen availability during high-altitude exercises, with saturation values often referred to as oxygen saturation. Oxygen saturation is defined as the percentage of hemoglobin in the blood that is carrying oxygen (Collins et al., 2015). As individuals ascend to higher altitudes, the partial pressure of oxygen decreases, leading to a reduction in oxygen saturation levels. It is important to note that responses to high-altitude exercise and saturation levels can vary significantly among individuals due to factors such as genetic predisposition, fitness level, training altitude, and acclimatization time (Siebenmann et al., 2012; Elmas & Elmas, 2020). Furthermore, altitude-related conditions such as acute mountain sickness or high-altitude pulmonary edema can further affect saturation levels.

Regarding heart rate measurements, significant differences were observed in pulse values recorded at the 3rd, 6th, 9th, and 12th minutes. Bahenský & Grosicki (2021) investigated the effects of high-altitude training on heart rate variability (HRV) in young athletes. The authors reported that high altitude training led to changes in HRV and autonomic modulation of heart rate. Specifically, they observed a decrease in parasympathetic activity and an increase in sympathetic activity, indicating a general shift towards sympathetic dominance. These changes in HRV may reflect adaptations in cardiovascular regulation in response to the hypoxic environment at high altitudes. Saunders et al. (2009) investigated the heart rate response to submaximal exercise at different altitudes in elite runners. They found that as altitude increased, there was a progressive increase in heart rate during submaximal exercise. This elevated heart rate response can be attributed to the reduced availability of oxygen at higher altitudes, leading to an increased sympathetic drive as an attempt to compensate for the reduced oxygen supply. These findings suggest that heart rate may serve as an indicator of the physiological strain imposed by high-altitude exercise. Similarly, Chapman et al. (2014) conducted a study examining the effects of altitude training on heart rate in athletes. They observed an increase in heart rate during exercise at higher altitudes compared to sea level. Additionally, they noted that heart rate recovery after exercise was slower at higher altitudes, indicating a prolonged sympathetic response. These findings further support the idea that high-altitude exercise causes changes in heart rate dynamics, potentially due to hypoxic stress imposed on the cardiovascular system.

Javaloyes et al. (2021) investigated HRV in trained cyclists during altitude training. They reported changes in HRV patterns, with decreases in both time domain and frequency domain measurements of HRV. These changes indicate a decrease in parasympathetic activity and an alteration in sympathovagal balance. They suggested that these adaptations in heart rate dynamics reflect the body’s physiological response to hypoxic stress and the increased demand on the cardiovascular system during high-altitude exercise. These studies demonstrate that high-altitude exercise elicits specific heart rate responses in athletes. The results of our study show that as altitude increases, pulse values also increase. The observed increases in heart rate during exercise, along with changes in HRV, suggest adaptations in sympathetic and parasympathetic cardiovascular control mechanisms. This may be due to reduced oxygen availability and the physiological adjustments required to meet the increased oxygen demand during high-altitude exercise. Heart rate responses observed during high-altitude exercises are important indicators of the physiological strain and adaptation processes that occur in athletes training at high altitudes (Bonato, Goodman & Tjh, 2023). Monitoring heart rate during training sessions at different altitudes can provide valuable information about cardiovascular responses and help optimize training protocols for athletes in high-altitude environments (Yu et al., 2022; Feng et al., 2023).

Limitations

This study has certain limitations. First, all participants underwent the four altitude conditions in a fixed order, from sea level to the highest altitude. The lack of a randomized or counterbalanced order may introduce potential order and training effects, which could have been minimized with a random order. However, our study aimed to observe altitude adaptation in a natural progression. Future studies may benefit from employing a randomized order to mitigate these potential effects. Additionally, our study assessed oxidative stress and muscle damage markers only immediately post-exercise. However, certain biomarkers, such as CK and MDA, may exhibit a delayed response, peaking 24 to 48 h after exercise. Future studies should incorporate follow-up measurements at multiple time points to better capture the time-dependent changes in oxidative stress and muscle damage, providing a more comprehensive understanding of the physiological responses to high-altitude exercise. These biomarkers were chosen based on their reliability and relevance to hypoxic conditions; however, a broader range of markers, such as inflammatory or metabolic indicators, could provide a more comprehensive view of the physiological responses. Future research could include additional biomarkers to expand the understanding of hypoxic adaptation. Furthermore, participants in this study resided at an altitude of 1,700 m, which may have influenced their physiological responses compared to those at lower altitudes. To improve generalizability, future studies could consider recruiting participants residing at sea level or lower altitudes.

Future perspectives

Changing oxygen levels at altitudes causes muscle damage and changes in oxidative stress levels. In the conclusion of the study, it was determined that exercises performed in a hypoxic environment at different altitudes increased the TAS and reduced the OSI in athletes. However, the adaptation response and subsequent recovery processes that occur after high-altitude exposure appear to contribute to enhanced antioxidant defense systems. These findings emphasize the importance of maintaining redox balance during high altitude training and provide insight into the mechanisms underlying the physiological adaptations observed in athletes. However, there is a need for more comprehensive studies on the subject, especially on the molecular changes that occur in this process, with different protocols, taking into account performance markers.

Conclusion

In conclusion, it was determined that exercises performed in a hypoxic environment at different altitudes increased TAS and reduced the OSI in elite long-distance runners. These findings suggest that altitude training not only elicits specific cardiovascular adaptations but also enhances the body’s antioxidant defense mechanisms, thereby mitigating oxidative stress. This dual benefit underscores the potential of high-altitude training to improve athletic performance and overall physiological resilience. Monitoring and understanding these changes can help optimize training strategies, ensuring athletes achieve peak performance while maintaining health and well-being in high-altitude environments.

Supplemental Information

Supplemental Information 1 Raw data

The authors thank all the subjects who participated in this study.

Additional Information and Declarations

Competing Interests

Author Contributions

Human Ethics

Data Availability

The authors declare there are no competing interests.

Mücahit Sarikaya conceived and designed the experiments, performed the experiments, analyzed the data, prepared figures and/or tables, and approved the final draft.

Beyza Öğe conceived and designed the experiments, performed the experiments, analyzed the data, prepared figures and/or tables, authored or reviewed drafts of the article, and approved the final draft.

Nuri Mert Embiyaoğlu conceived and designed the experiments, analyzed the data, authored or reviewed drafts of the article, and approved the final draft.

Muzaffer Selçuk analyzed the data, authored or reviewed drafts of the article, and approved the final draft.

Vedat Çınar analyzed the data, prepared figures and/or tables, and approved the final draft.

Salih Öner analyzed the data, authored or reviewed drafts of the article, and approved the final draft.

Yıldırım Gökhan Gencer analyzed the data, authored or reviewed drafts of the article, and approved the final draft.

Mehdi Aslan analyzed the data, authored or reviewed drafts of the article, and approved the final draft.

Mustafa Sencer Ulema analyzed the data, authored or reviewed drafts of the article, and approved the final draft.

Yunus Emre Yarayan analyzed the data, authored or reviewed drafts of the article, and approved the final draft.

Kadir Keskin analyzed the data, authored or reviewed drafts of the article, and approved the final draft.

Nouf H. Alkhamees analyzed the data, authored or reviewed drafts of the article, acquisition of funding, and approved the final draft.

Bodor Bin Sheeha analyzed the data, prepared figures and/or tables, and approved the final draft.

Gerasimos V. Grivas analyzed the data, authored or reviewed drafts of the article, and approved the final draft.

Sameer Badri AL-Mhanna analyzed the data, prepared figures and/or tables, authored or reviewed drafts of the article, and approved the final draft.

Alexios Batrakoulis analyzed the data, authored or reviewed drafts of the article, and approved the final draft.

The following information was supplied relating to ethical approvals (i.e., approving body and any reference numbers):

This study was conducted in accordance with the Declaration of Helsinki and approved by the Ethics Committee of Van Yüzüncü Yıl University our institution (protocol code TYD-2021-9553 and date of approval 15 May 2021). The participants provided their written informed consent to participate in this study.

The following information was supplied regarding data availability:

The raw measurements are available in the Supplementary File.

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
