# Peer review of "The acute effects of simulated hypoxic training at different altitudes on oxidative stress and muscle damage in elite long-distance runners"

_PeerJ, doi:10.7717/peerj.19338_

## Round 0.1 · original submission · Major Revisions

Please carefully address reviewers' comments

Note the PDF from Reviewer 2

Reviewer 1 ·

Basic reporting

Thank you for the opportunity to review this manuscript entitled ‘The acute effects of simulated hypoxic training at different altitudes on oxidative stress and muscle damage in elite long-distance runners’ by Sarikaya and co-authors. In the present study, authors examined the acute responses to exercise on different simulated altitudes. I think that this study is important because understanding the mechanism of the hypoxic training, a common way to improve endurance performance in athletes, is essential to adapt the training stimuli to the athlete. I think that this article has some methodological doubts that need to be addressed before being considered for publication. Below you will find my comments.

Abstract: In my opinion the abstract misses a background and some practical applications of the results. I suggest rewriting the background, adding what is already known, and adding at least a sentence in the conclusions stating the practical applications of your results. Moreover, I suggest summarizing the results section, maintaining only the most relevant results.

Introduction: I think that this section well presents what is already known in literature about the topic of the article, but it misses a section about why their study is needed. Moreover the creatine kinase (CK) is introduced just at the end of the introduction, with the aim of the study. Try to explain why assessing CK is needed.

Experimental design

Materials and methods: I think that you need to explain more about the study design. How it can be considered “single-blinded” and “randomized”? What does it mean “balanced”?. At line 157 you state “exercise sessions were conducted in a Hypoxico Everest Summit II Altitude Generator”, however it is not clear if it was performed in a tent or using a mask. It is not clear if the four exercise sessions were performed all on the same day or in different days. This could really influence the results of the study. Please delete the description of the hypoxico generator from line 183 to 187. Why did you choose 1700m if the athletes lived at 1700m? Why did you use the d of cohen instead of the ηp2 to assess the effect size?

Validity of the findings

Results: I think that this section is too long and most of the information’s the authors reports can be found in the tables. I suggest reporting only the relevant differences between repeating the numerical information that can be found in the tables. You also forgot to place Figure 1 in the text.

Discussion: Please rephrase this section. In my view, the authors present the results of various studies; however, when these findings contrast with those of other studies, they do not attempt to explore potential explanations for these discrepancies, which could be attributed to differences in study protocols.

·

Basic reporting

The article is interesting as it attempts to address many of the questions regarding the acute effects of hypoxia. However, it has significant limitations that the authors should address. It is difficult to follow, lacks references throughout the document, and the information is presented in a disorganized and repetitive manner across different sections. I have attached specific comments throughout the document.

Experimental design

The design is appropriate and the objective is interesting. However, it should be rewritten and explained more clearly in the introduction and methods sections.

Validity of the findings

The results should be explained more clearly, and repetitive information should be avoided.

---

## Round 0.2 · accepted · Accept

I confirm that all reviewer comments have been thoroughly addressed by the authors. As the previous reviewer 2 was unable to assess the revised manuscript, I have reviewed it myself. I now consider the manuscript ready for publication.

Reviewer 1 ·

Basic reporting

I believe the authors have thoroughly addressed my previous comments.

Experimental design

I believe the authors have thoroughly addressed my previous comments.

Validity of the findings

I believe the authors have thoroughly addressed my previous comments.

Additional comments

I believe the authors have thoroughly addressed my previous comments.